

# Instrumental validity and intra/inter-rater reliability of a novel low-cost digital pressure algometer

Daniel Jerez-Mayorga[1], Carolina Fernanda dos Anjos[2],
Maria de Cássia Macedo[2], Ilha Gonçalves Fernandes[2],
Esteban Aedo-Muñoz[3], Leonardo Intelangelo[4] and
Alexandre Carvalho Barbosa[2]

[1] Facultad de Ciencias de la Rehabilitación, Universidad Andrés Bello, Santiago, Santiago, Chile
[2] Department of Physical Therapy, Federal University of Juiz de Fora, Governador Valadares, Minas Gerais, Brazil
[3] Department of Physical Education, Federal University of Juiz de Fora, Governador Valadares, Minas Gerais, Brazil
[4] Department of Physical Therapy, Universidad del Gran Rosario, Rosario, Santa Fe, Argentina

Corresponding author
Alexandre Carvalho Barbosa,
alexandre.barbosa@ufjf.edu.br

## ABSTRACT

**Background:** Pain assessment is a key measure that accompanies treatments in a wide range of clinical settings. A low-cost valid and reliable pressure algometer would allow objective assessment of pressure pain to assist a variety of health professionals. However, the pressure algometer is often expensive, which limits its daily use in both clinical and research settings.

**Objectives:** This study aimed to assess the instrumental validity, and the intra- and inter-rater reliability of an inexpensive digital adapted pressure algometer.

**Methods:** A single rater applied 60 random compressions on a force platform. The pressure pain thresholds of 20 volunteers were collected twice (3 days apart) by two raters. The main outcome measurements were as follows: the maximal peak force (in kPa) and the pressure pain threshold (adapted pressure algometer vs. force platform). Cronbach's $\alpha$ test was used to assess internal consistency. The standard error of measurement provided estimates of measurement error, and the measurement bias was estimated with the Bland–Altman method, with lower and upper limits of agreement.

**Results:** No differences were observed when comparing the compression results ($P = 0.51$). The validity and internal intra-rater consistencies ranged from 0.84 to 0.99, and the standard error of measurement from 0.005 to 0.04 kPa. Very strong ($r = 0.73–0.74$) to near-perfect ($r = 0.99$) correlations were found, with a low risk of bias for all measurements. The results demonstrated the validity and intra-rater reliability of the digitally adapted pressure algometer. Inter-rater reliability results were moderate ($r = 0.55–0.60$; Cronbach's $\alpha = 0.71–0.75$).

**Conclusion:** The adapted pressure algometer provide valid and reliable measurements of pressure pain threshold. The results support more widespread use of the pressure pain threshold method among clinicians.

# INTRODUCTION

Pain has been described as a multidimensional construct involving psychological and physical domains with different patterns depending on the emotional state (*Melia et al., 2015*). These characteristics may impair conclusions and lead to biased clinical reasoning regarding the patterns of group pain due to intra-group and longitudinal variability in subjects' co-morbidities and momentaneous emotional state. Nevertheless, physical assessment is essential to provide objective data to compare the prospective effects of interventions for pain management (*Imamura et al., 2013*; *Calixtre et al., 2016*; *Intelangelo, Bordachar & Barbosa, 2016*).

Pain is mostly assessed by patient self-reports using the visual analog scale (*Walton et al., 2011*). Self-reported pain intensity is important and reflects physiological and psychological features. However, it can be difficult to interpret because of its subjectivity and overestimation of pain level (*González-Fernández et al., 2014*). Objective pain assessment is essential to establish a prospective evaluation, compare baseline results to other timeline assessments, or even as a prognostic measure to predict future outcomes (*Walton et al., 2011*, *2013*). Thus, pressure algometry is a diagnostic aid used to assess some musculoskeletal problems. The pressure pain threshold has been used to aid the diagnosis of pain by providing a quantified force value of tissue tenderness and occurs at the minimum transition point when the applied pressure is sensed as pain (*Kinser, Sands & Stone, 2009*; *Cunha et al., 2014*). The pressure algometer is an instrument used to assess the pressure pain threshold for both regional and widespread musculoskeletal pain (*Durga et al., 2016*). This equipment includes a system to convert the force applied through a 1 cm$^2$ pressure application surface to Newtons (N/cm$^2$) or kilograms of force (kgf/cm$^2$), and a display. The units can be easily converted to kilopascal (kPa), the international metric for pressure (1 kg/cm$^2$ = 98.066 kPa). The pressure algometer enables the rater to semi-objectively quantify the mechanical sensitivity to pain level and the recovery of underlying problems or soreness levels (*Kinser, Sands & Stone, 2009*; *Alburquerque-Sendín et al., 2018*).

The instrumental validity of commercial pressure algometers has already been assessed in previous studies. *Kinser, Sands & Stone (2009)* and *Vaughan, McLaughlin & Gosling (2007)* manually applied pressure on a force platform to test the reliability and construct validity of pressure algometers. Both studies found high levels of correlation between the force platform and pressure algometers. Other studies have assessed the responsiveness of a pressure algometer to diagnose dysfunctional conditions. *Ko et al. (2016)* assessed the correlation between a modified pressure algometer and a commercial algometer to assess the pressure pain threshold of the epigastric region. Unfortunately, commercially available pressure algometers are expensive and may require specific software for reporting and viewing the results, resulting in more time and training required to assess the

pressure pain threshold. The validation of an easy-to-read, low-cost, digitally adaptable pressure algometer would enable widespread quantitative measurements of pressure pain thresholds in clinical practice routine, benefiting early assessment of pain conditions in low-income and developing countries, mainly in primary care (*Kinser, Sands & Stone, 2009*; *O'Connor, Baweja & Goble, 2016*). A portable pressure algometer adapted from a hanging scale may be a cost-effective alternative to ensure accurate algometry assessments.

The hanging scale is a battery-operated instrument used to weigh objects suspended from an attachment. The equipment uses a load cell, which is a metallic sturdy element, yet elastic enough for a load to deform it. The load cell is attached to a strain gauge, which reads the change in electrical resistance when a pressure or traction load is placed in the load cell. The change in electrical resistance is converted to a digital signal by the strain gauge, and the result appears on a display (*Hanafee & Radcliffe, 1967*). Among other measures, the correlation level between the result of a certain instrument and some external criterion must be confirmed to determine the instrumental validity of an equipment. The criterion has to be a widely accepted measure and considered as the gold-standard method, with the same measurement characteristics of the assessment tool (*De Souza, Alexandre & De Guirardello, 2017*; *Sullivan, 2011*).

The purpose of this study was to examine the instrumental validity as well as the intra- and inter-rater reliability of a low-cost pressure algometer adapted from a hanging scale. Validity was assessed by comparing differences in the measurements of a series of random peak forces applied on a laboratory-grade force platform. Force platforms measure vertical ground reaction forces in response to compressions applied on the surface. They are considered as the gold-standard devices for ground reaction forces owing to their high measurement precision (*O'Connor, Baweja & Goble, 2016*; *Carlos-Vivas et al., 2018*). The hypothesis is that a low-cost adapted pressure algometer is valid and reliable to be considered an acceptable method to assess the pressure pain threshold.

## MATERIALS AND METHODS

### Equipment

All data were collected at the facilities of the Clinic-School of Physical Therapy, Federal University of Juiz de Fora, in May 2019. The low-cost adapted pressure algometer (MED.DOR Ltd., Brazil; maximum compression = 50 kgf, precision = 0.1 kgf, 3-digit display) had a 5-cm screw attached to the distal extremity. A 1-cm$^2$ round rubber application surface was attached to follow the standardization for pressure algometry (Fig. 1). The low-cost adapted pressure algometer calibration is checked by placing a known weight (1 kg) on the application surface. The maximal tolerated difference between the weight and the value on the display is 0.1 kgf. The adapted algometer used in the present study was brand new, and the calibration was checked twice before any measurement.

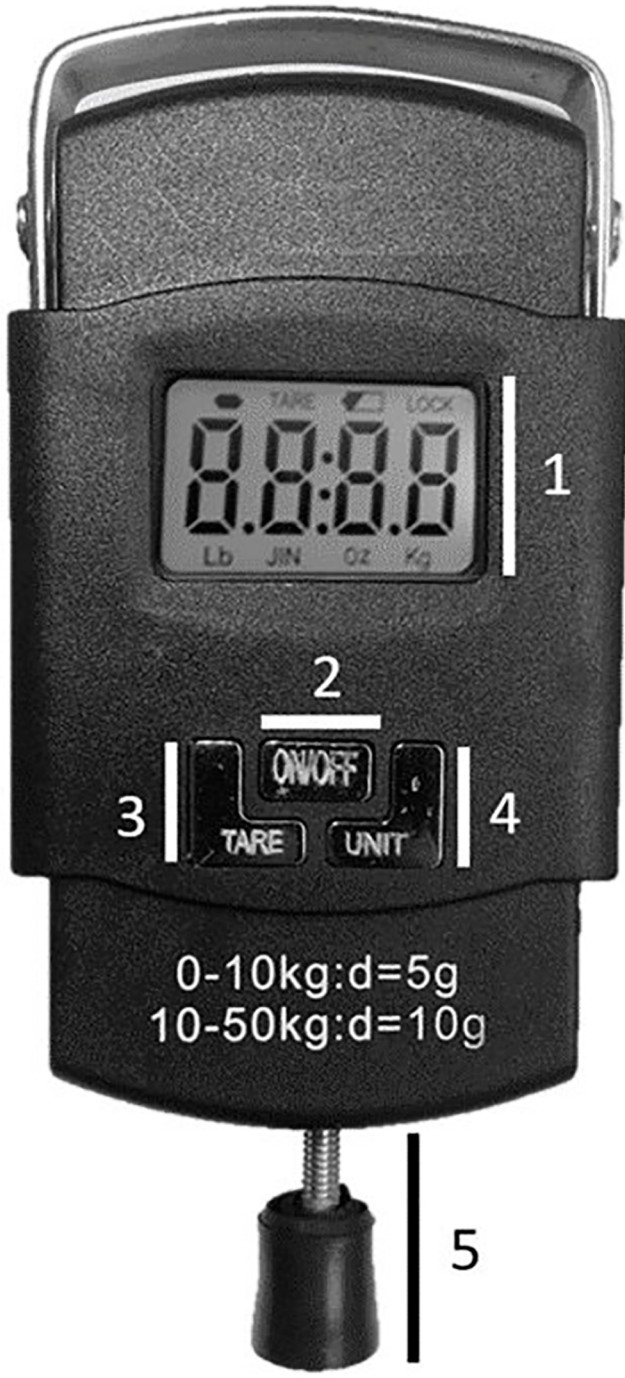

**Figure 1 Adapted pressure algometer—PA.** (1) Display; (2) On-Off button; (3) Tare button; (4) Unit selection button; (5) Adapted terminal.  

A two-axis force platform (37 cm × 37 cm; Pasport PS-2142; PASCO, Roseville, CA, USA) collected data using five force beams (sample rate = 1,000 Hz). Four beams in the corner were used to measure the vertical force (range: −1,100 N to +4,400 N) and a 5th beam measured the force in a parallel axis (range: −1,100 N to +1,100 N). The recorded trials were converted to kPa (1 kg/cm$^2$ = 98.066 kPa).

## Procedures

### Instrumental validity

An independent rater performed 60 random 3-s pressure trials using the adapted pressure algometer on the force platform with an interval of 3 s. Data were collected and stored using the PASCO Capstone Software (Version 1.13.4; PASCO Scientific, 2019), and the low-cost adapted pressure algometer display readings were recorded using an off-board USB synchronized camera.

### Intra-and inter-rater reliability

Before the assessment of any participant, two independent raters performed a 4-day training protocol. The protocol consisted of applying constant-progressive pressure with a low-cost adapted pressure algometer on a laboratory-grade load cell (Miotec™ Biomedical Equipment, Porto Alegre, RS, Brazil; maximum tension-compression = 200 kgf, precision = 0.1 kgf, maximum error = measurement = 0.33%) with Miotec™ software for visual feedback (MioTrainer™ Biomedical Equipment, Porto Alegre, RS, Brazil) for two nonconsecutive days (3 nonconsecutive hours per day). The conversion from analog to digital signals was performed by an A/D board (Miotec™ Biomedical Equipment, Porto Alegre, RS, Brazil) with a 16-bit resolution input range, a sampling frequency of 2 kHz, a common rejection module greater than 100 dB, a signal-noise ratio less than 03 μV, root mean square, and impedance of 109 Ω. All data were recorded and processed using the Miotec Suite™ software (Miotec™ Biomedical Equipment, Porto Alegre, RS, Brazil). An assessor monitored the pressure applied by the raters for two consecutive days using the same software, but the raters did not receive any visual feedback. The training was aimed at ensuring the velocity to apply pressure using a low-cost adapted algometer (1 kg/s).

The independently trained raters collected the middle deltoid muscle's pressure pain threshold of 20 participants (10 women; 22 ± 2 years; 63 ± 13 kg; 160 ± 10 cm; 23 ± 4 kg/cm$^2$). The exclusion criteria for participants included: body mass index >28 kg/cm$^2$, any self-reported health issues, alcohol consumption within 5 days prior to the assessments, shoulder pain, previous shoulder surgery, or any diagnosed shoulder or cervical impairment. The objectives of the study were explained to the subjects, who were notified of the benefits and potential risks involved before signing an informed consent form prior to participation. The Federal University of Juiz de Fora ethics committee for human investigation approved the procedures employed in the study (reference number: 02599418.9.0000.5147).

The pressure pain threshold was collected twice (3 days apart: days 1 and 2). To evaluate the intra- and inter-rater reliability, the following positioning was adopted: (1) the participant remained seated with the feet on the floor, (2) the hands rested on the thighs, and (3) the trunk was erect. The middle deltoid's site received progressive 1 kg/s pressure controlled by a metronome until the participant experienced pain (*Kinser, Sands & Stone, 2009*). An effort was made to standardize the anatomic locations of each session. The same rater was responsible for palpating and marking the pressure pain threshold site on each subject before any measurements, both on days 1 and 2. The middle
deltoid's site was topographically determined in the middle of a horizontal line drawn between the acromioclavicular joint and the deltoid muscle insertion (*Ribeiro et al., 2016*). Three measurements were performed for each site, with 10–15 s apart. The first measurement was discarded (*Ylinen et al., 2007*; *Nussbaum & Downes, 1998*).

The participant lifted the opposite hand when the pressure pain threshold was achieved, that is, when the applied pressed evoked pain. The examiner pressured the "tare" button to lock the reading, immediately retracting the adapted pressure algometer. Then, the pressure pain threshold reading was registered (*Wytrążek et al., 2015*).

## Statistical analysis

The recorded peaks were then extracted. All trials were used for analysis, consisting of the following: a total of (1) 60 measurements (validity analysis, force platform vs. adapted pressure algometer) and (2) 80 measurements (reliability analysis). Data are presented as mean values and standard deviations. The independent Student's *t*-test was used to compare differences between measurements in the validation process. The intra- and inter-rater differences were compared using the mixed between- (rater 1 vs. rater 2) and within-subject analysis (moment and moment*rater) of variance with repeated measures. All data were reworked using Holm's post hoc test to avoid multiple testing. Significance was set at $p < 0.05$. Intraclass correlation coefficients ($ICC_{(2,1)}$) were calculated to compare the results between both types of equipment and raters. Poor reliability was indicated by values less than 0.5, moderate reliability between 0.5 to 0.75, good reliability between 0.75 and 0.9, and excellent reliability greater than 0.90 (*Koo & Li, 2016*). Chronbach's α test was used to assess the expected correlation of both types of equipment measuring the same construct. The standard error of measurement (SEM) was also calculated to provide an estimate of measurement error. A linear regression was used to estimate the coefficient of correlation ($r$) and the adjusted coefficient of determination ($r^2$). The magnitude of the correlation was qualitatively interpreted using the following thresholds: <0.1, trivial; 0.1–0.3, small; 0.3–0.5, moderate; 0.5–0.7, large; 0.7–0.9, very large; and >0.9, nearly perfect (*Hopkins et al., 2009*). The Bland–Altman method estimated the measurement bias, with lower and upper limits of agreement between results. Statistical analyses were performed using Jamovi software (Jamovi project, version 0.9, 2018).

## RESULTS

### Validity: Force platform vs. PA

No significant differences were observed in pressure trials (Table 1) between the adapted pressure algometer (405.63 ± 235.34 kPa) and the force platform (434.15 ± 239.18 kPa; $p = 0.25$). The $ICC_{(2,1)}$ and Cronbach's α returned values of 0.98 and 0.99, respectively. The SEM returned a value of 0.005 kgf, and the linear regression showed statistically significant results ($r = 0.99$; adjusted $r^2 = 0.99$; $p = 0.001$). The Bland–Altman results showed high levels of agreement (Fig. 2).

**Table 1 Validity, intra-and inter-rater reliability pairwise comparisons.**

| Type of Analysis | Outcome | $p_{holm}$ | Mean difference | 95% Confidence Interval | |
|---|---|---|---|---|---|
| | | | | Lower | Upper |
| Validity | Adapted Pressure Algometer vs. Force Platform | 0.512 | −28.5 | −114 | 57.3 |
| Intra-rater | Rater 1 - Moment 1 vs. Moment 2 | 0.83 | −2.55 | −27.3 | 22.2 |
| | Rater 2 - Moment 1 vs. Moment 2 | 0.93 | −1.03 | −25.9 | 23.9 |
| Inter-rater | Moment 1 - Rater 1 vs. Rater 2 | 0.65 | −10.79 | −58.1 | 36.5 |
| | Moment 2 - Rater 1 vs. Rater 2 | 0.68 | −9.27 | −54.5 | 35.9 |

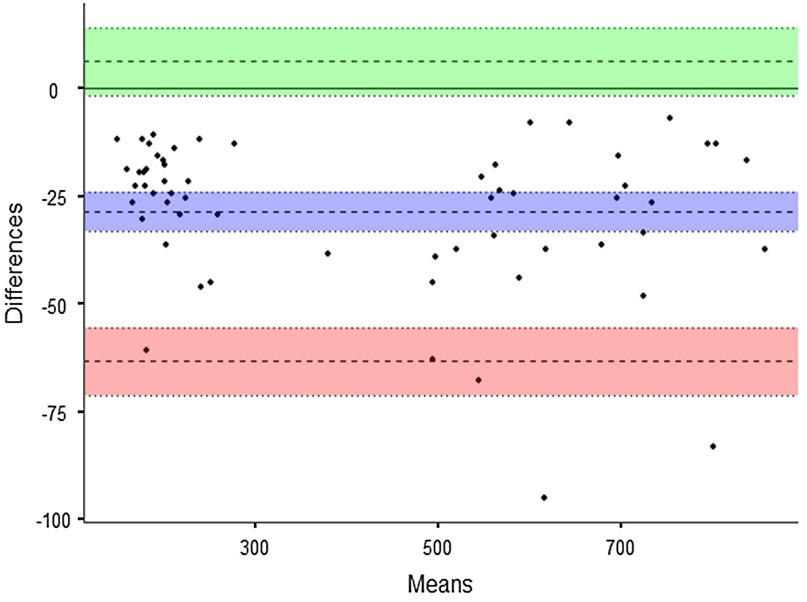

**Figure 2 Bland-Altman plot: instrumental validity.** Bias = −28.52 (95% confidence interval (CI) [−33.10 to −23.9]); lower limit of agreement (LLA) = −63.29 (95% CI [−71.17 to −55.4]); upper limit of agreement (ULA) = 6.25 (95% CI [−1.62 to 14.1]).

## Intra- and inter-rater reliability

The pressure pain threshold from both raters showed very low variation over time (Rater 1: Day 1 = 203 ± 74 kPa, Day 2 = 206 ± 71.6 kPa; Rater 2: Day 1 = 214 ± 73.7 kPa, Day 2 = 215 ± 69.6 kPa). The intra-rater comparison showed no significant differences (Moment: $F = 0.05$; $p = 0.83$ and Moment*Rater: $F = 0.01$; $p = 0.93$) (Table 1). The $ICC_{(2,1)}$ and Cronbach's α analysis returned relevant values (Rater 1: $ICC_{(2,1)} = 0.76$, Cronbach's α = 0.85; Rater 2: $ICC_{(2,1)} = 0.73$, Cronbach's α = 0.84). The SEM values were low (Rater 1 = 0.02, Rater 2 = 0.01), and moderate values were also obtained in the linear regression analysis (Rater 1: $r = 0.74$, adjusted $r^2 = 0.52$; Rater 2: $r = 0.73$, adjusted $r^2 = 0.50$). The Bland–Altman results showed high levels of agreement (Fig. 3).
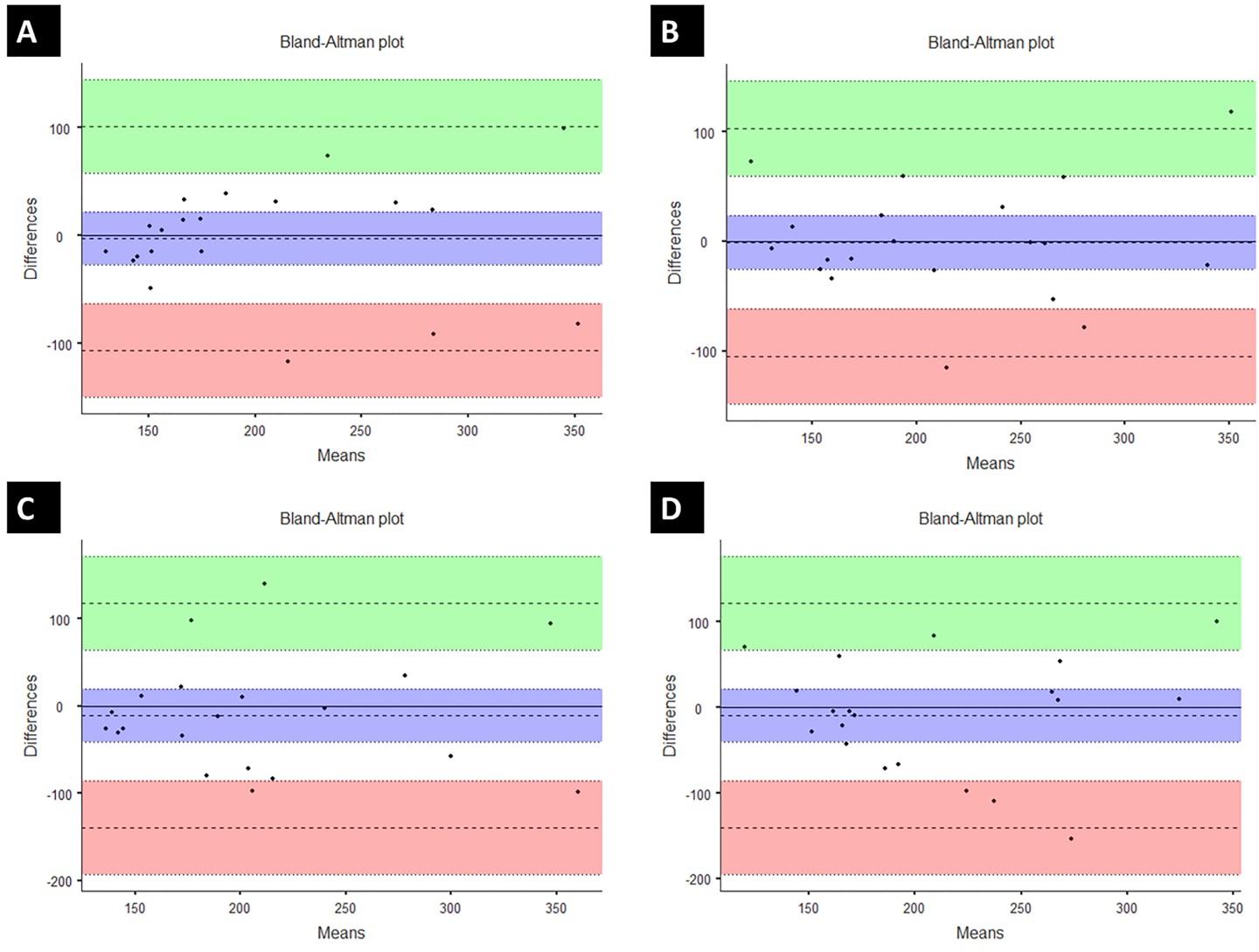

**Figure 3 Bland-Altman plot: intra-rater reliability.** (A) Rater 1: Bias = −2.55 (95% confidence interval (CI) [−24.3 to 22.2]); lower limit of agreement (LLA) = −106.38 (95% CI [−149 to −63.3]); upper limit of agreement (ULA) = 101.28 (95% CI [58.2–144.4]). (2) (B) Rater 2: Bias = −1.03 (95% CI [−25.9 to 23.9]); LLA = −105.28 (95% CI [−148.6 to −62]); ULA = 103.22 (95% CI [59.9–146]). Inter-rater reliability. (C) Day 1: Bias = −10.8 (95% CI [−41.6 to 20]); LLA = −139.8 (95% CI [−193.4 to −86.3]); ULA = 118.2 (95% CI [64.7–171.8]); (D) Day 2: Bias = −9.27 (95% CI [−40.7 to 22.1]); LLA = −140.77 (95% CI [−195.4 to −86.2]); ULA = 122.23 (95% CI [67.6–176.8]).

The inter-rater reliability showed no differences among measurements ($F = 0.22$; $p = 0.64$) (Table 1), with moderate results for reliability analysis (Day 1: $ICC_{(2,1)} = 0.56$, Cronbach's $\alpha = 0.75$; Day 2: $ICC_{(2,1)} = 0.54$, Cronbach's $\alpha = 0.71$). The SEM results showed very low values (Day 1: 0.04 kgf, Day 2: 0.02 kgf), and moderate values in the linear regression analysis (Day 1: $r = 0.60$, adjusted $r^2 = 0.33$; Day 2: $r = 0.55$, adjusted $r^2 = 0.26$). The Bland–Altman analysis showed acceptable levels of agreement (Fig. 3).

## DISCUSSION

This study was designed to examine the instrumental validity and the intra- and inter-rater reliability of an adapted low-cost pressure algometer. The results showed no significant

differences in the peak compressive force recorded from the adapted pressure algometer and the force platform. A significant correlation was observed between the low-cost adapted algometer and the force platform while measuring the same construct. These findings support the primary hypothesis, which contends that a low-cost pressure algometer is both valid and reliable to be considered as a standard equipment to assess the pressure pain threshold. Therefore, the tested device seems to be an acceptable alternative to expensive instruments.

Previous studies showed acceptable levels of validity and reliability of other digital algometry systems (*Kinser, Sands & Stone, 2009*; *Durga et al., 2016*). *Kinser, Sands & Stone (2009)* tested the construct validity of a digital pressure algometer using the same protocol as our study by manually applying pressure on a force platform. The authors used ten sets of five applications to 80 N and one additional set of five applications to subsequent levels of progressive 10 N (20, 30, 40, 50, 60, 70, 80, 90, 100, and 110 N). The results showed high correlation between the tested algometer and the force platform ($r = 0.99$) for both 80 N and incremental trials. *Vaughan, McLaughlin & Gosling (2007)* also used the force platform as a reference instrument to test the validity of a digital pressure algometer. The authors applied 300 vertical pressures on the force platform with progressive pressure rates (10, 20, 30, 40, and 50 kPa/s). The result showed an excellent ICC range (0.90–0.99) for all comparisons. In general, all previous studies and the present work suggest the validity of the results obtained from a digital pressure algometer. These excellent results could be attributed to the strain gauge-based system used to acquire the signals. The conversion from analogic (load cell deformation) to electrical-digital signal (strain gauge) is very effective, even in very affordable systems. As the resistance varies in a sturdy element with the applied force, the strain gauge converts the force (in this case, pressure) into a change in electrical resistance that can be measured.

Several studies have evaluated the reliability of distinct pressure algometers as a tool to distinguish healthy individuals from those with musculoskeletal disorders. *Balaguier, Madeleine & Vuillerme (2016)* found high reliability between all three pressure pain threshold measures at sites in the lower back. *Walton et al. (2011)* assessed the intra-rater and inter-rater reliability of an accessible digital algometer in 60 healthy volunteers and 40 individuals with neck pain. The authors tested the upper fibers of the trapezius and tibialis anterior muscles. The intra-rater ICC results in both groups ranged from 0.94 to 0.97 for the trapezius and tibialis anterior muscles. The inter-rater ICC range (0.79–0.90) was lower than that of the intra-rater due to variations between observers. However, both results were considered adequate. *Waller et al. (2015)* found high intra- and inter-rater reliability (ICC = 0.81–0.99; ICC = 0.92–0.95, respectively) using five research assistants. Each assistant tested 20 pain-free subjects at the wrist, leg, cervical, and lumbar spine. The intra-rater SEM ranged between 79 and 100 kPa. However, *Van Wilgen, Van der Noord & Zwerver (2011)* found lower values for intra-rater reliability compared to inter-rater reliability of pressure algometry in healthy volleyball athletes and those with patellar tendinopathy. The authors found high inter-rater reliability (ICC = 0.93), but only moderate intra-rater reliability (ICC = 0.60) for pain pressure threshold measurements. The authors argued that the lower intra-rater ICC values were

probably due to variance within the observer and also within the athletes, as the pain in patellar tendinopathy varies over time. The diagnosis of fibromyalgia utilizes the pressure pain threshold as a key assessment to distinguish healthy individuals from those with fibromyalgia (*Gómez-Perretta et al., 2016*; *Cheatham et al., 2018*). Neck pain, cranio-cervical headache, and temporomandibular disorders also include the pressure pain threshold as an important component for clinical reasoning about the level of severity, influencing the treatment direction (*Walton et al., 2011*; *Cunha et al., 2014*; *Castien, Van der Wouden & De Hertogh, 2018*).

However, those previous studies used commercial pressure algometers. For clinical and ambulatory settings, the high cost and the user's interface would be an issue to obtain fast objective pain measurements, requiring both training and experience for assessments. Brazilian physiotherapists have an average monthly salary of USD 500, according to the Occupational Brazilian Classification (https://www.salario.com.br/profissao/fisioterapeuta-geral-cbo-223605/). The adapted pressure algometer used in this study had a production cost of USD 10.00, while the standard digital equipment cost ranged from USD 600.00 to USD 1,000.00. The validation procedure enables use of the low-cost adapted pressure algometer for clinical assessments in a practice routine, which may directly impact primary and ambulatory care in low-income and developing countries, by adding an objective and inexpensive tool to assess the pressure pain threshold.

Some limitations of the present study must be addressed. The pressure pain threshold in body sites other than the deltoid muscle must be assessed to ensure the validity of the adapted pressure algometer on different sites. However, we hypothesize that they should not give any different results to direct assessment using the adapted pressure algometer, since the standard deviation remained at very low values and the current results gave very good measures compared to the force platform and additional good reliability. The instrumental validity of an equipment's measurements also ensures unbiased assessments (*Gadotti, Vieira & Dj, 2006*).

Other studies have identified different factors to consider when evaluating the pressure pain threshold, such as gender and obesity (*Chesterton et al., 2003*; *Price et al., 2013*). A review of studies involving induced pain found a consistent pattern of women exhibiting greater pain sensitivity and a reduction in pain inhibition compared to men (*Bartley & Fillingim, 2013*). In addition, the characteristic of pain imposed is an important factor for these differences, since the type of pressure pain has one of the highest effect sizes in the pain report (*Nussbaum & Downes, 1998*; *Robinson et al., 1998*). It is suggested that interactions between biological and psychosocial factors are responsible for these gender differences, but all studies indicate the need for additional research to elucidate the mechanisms that drive gender differences in pain responses (*Chesterton et al., 2003*; *Bartley & Fillingim, 2013*; *Robinson et al., 1998*). Some studies suggest that in areas with additional subcutaneous fat, pain thresholds for electrical or pressure stimuli increase and pain sensitivity decreases in obese individuals (*Price et al., 2013*; *Khimich, 1997*). A study has also shown biochemical changes in trigger points with higher levels of inflammatory mediators, catecholamines, and cytokines in obese individuals

(*Shah et al., 2008*). Mechanical stretching of the skin in response to excess fat can lead to a decrease in the density of nociceptive fibers, and obesity is associated with the chemical inhibition of pain with an increase in β endorphin and endogenous opioid peptide (*Price et al., 2013*). The present study had a balanced cohort with regard to participant sex, and all participants were classified as normal according to their body mass index. However, the current sample was chosen only for reliability analysis. Pressure pain threshold as a clinical result is well established, but more studies should take into account sex and body mass index differences to avoid bias in experimental protocols (*Chesterton et al., 2003*). Pressure pain threshold was also positively but poorly correlated with high-density lipoprotein cholesterol (*Zhang et al., 2013*). A high pressure pain threshold was also found among subjects with hyperglycemia and excessive alcohol consumption (*Zhang et al., 2013*). In the present study, no blood assessment was performed to exclude those factors. However, the sample consisted of young adults, decreasing the chance of any important health issues. Additionally, exclusion criteria included previous excessive alcohol consumption.

Considering its portability, easy assembly, and lower cost, the currently tested device seems to be a valid standard equipment for pressure pain threshold assessment. Therefore, the adapted pressure algometer is a valid device providing similar measurements compared to a force platform. The portability, cost-effectiveness, and friendly user system provide an effective way to measure the pressure pain threshold.

## CONCLUSIONS

The current hypothesis is that a low-cost pressure algometer is valid and reliable enough to be considered as a standard equipment to assess the pressure pain threshold. The results showed that the low-cost adapted pressure algometer is a valid tool compared to a force platform. The low-cost adapted pressure algometer is also reliable for assessing the pressure pain threshold. Future directions include evaluating the low-cost adapted pressure algometer in routine clinical assessments for the systematic evaluation of pressure pain. Further studies should consider other assessments, such as temporal summation and conditioned modulated pain, using a low-cost adapted pressure algometer.

## ACKNOWLEDGEMENTS

Special thanks to the UFJF-GV Department of Physical Therapy and to the Dean of International Relations—UFJF. We would like to thank Editage for English language editing.

### Funding

This study was financed by Coordenação de Aperfeiçoamento de Pessoal de Nível Superior—Brasil (CAPES)—Finance Code 001, and by Fundação de Amparo à Pesquisa de Minas Gerais (FAPEMIG)—Finance Code APQ-02040-18. The funders had no role in study design, data collection and analysis, decision to publish, or preparation of the manuscript.

## Grant Disclosures

The following grant information was disclosed by the authors:
Coordenação de Aperfeiçoamento de Pessoal de Nível Superior—Brasil (CAPES): 001.
Fundação de Amparo à Pesquisa de Minas Gerais (FAPEMIG): APQ-02040-18.

## Competing Interests

The authors declare that they have no competing interests.

## Author Contributions

- Daniel Jerez-Mayorga analyzed the data, authored or reviewed drafts of the paper, and approved the final draft.
- Carolina Fernanda dos Anjos conceived and designed the experiments, prepared figures and/or tables, authored or reviewed drafts of the paper, and approved the final draft.
- Maria de Cássia Macedo conceived and designed the experiments, performed the experiments, prepared figures and/or tables, and approved the final draft.
- Ilha Gonçalves Fernandes conceived and designed the experiments, performed the experiments, prepared figures and/or tables, authored or reviewed drafts of the paper, and approved the final draft.
- Esteban Aedo-Muñoz performed the experiments, analyzed the data, authored or reviewed drafts of the paper, and approved the final draft.
- Leonardo Intelangelo analyzed the data, authored or reviewed drafts of the paper, and approved the final draft.
- Alexandre Carvalho Barbosa conceived and designed the experiments, performed the experiments, analyzed the data, authored or reviewed drafts of the paper, and approved the final draft.

## Human Ethics

The following information was supplied relating to ethical approvals (i.e., approving body and any reference numbers):

Federal University of Juiz de Fora ethics committee for human investigation approved this study (02599418.9.0000.5147).

## Data Availability

The raw measurements are available as a Supplemental File.

## Supplemental Information

Supplemental information for this article can be found online at http://dx.doi.org/10.7717/peerj.10162#supplemental-information.

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
