# Peer review of "Instrumental validity and intra/inter-rater reliability of a novel low-cost digital pressure algometer"

_PeerJ, doi:10.7717/peerj.10162_

## Round 0.1 · original submission · Major Revisions

Thank you for your submission. The reviewers have recommended a major revision of the manuscript and have made specific recommendations. Please address each of these points in your revised manuscript and include an itemized reply to each reviewer.

·

Basic reporting

See below

Experimental design

See below

Validity of the findings

See below

Additional comments

Manuscript Number: PeerJ 46274
Title: Instrumental validity and intra/inter-rater reliability of a novel low-cost digital pressure algometer

Main comments
This study aimed to assess the instrumental validity, the intra- and inter-rater reliability of an inexpensive digital adapted pressure algometer. The aim is of interest for clinical research related to musculoskeletal disorders.
The paper is generally well written and of interest. There are few flaws that need to be addressed in a revision and this is definitely feasible. It is erroneous to report pressure in Kg or N – the authors should comply with the SI meaning that pressure is reported in Pa (or kPa). Then all results related force expressed in Kg or N needs to be converted into Pa (see e.g. https://www.convertunits.com/from/kg/cm2/to/kpa ). Make also changes elsewhere (abstract, introduction and discussion). More the choice of force platform and load cell should be justified better. Finally, it would be good how the re-calibration should be done so potential drift does not influence PPT collected longitudinally.
See below for other suggestions for improving the manuscript. Reduce evt the number of abbreviations.

Specific comments
In general, make clear if PA relates to the expensive or cheap algometer.
Line 52-53 “quantify the pain level and the recovery of underlying problems or soreness levels” PA does not quantify pain levels – it is more appropriate to write “mechanical sensitivity to pain” see a recent PPT review https://www.ncbi.nlm.nih.gov/pubmed/29403305. It probably better to refer PPT as a semi-objective method (the patient still reports when pressure turns into pain)
Line 54 define expensive/inexpensive
Paragraph equipment
It is a bit unclear why both force platform and load cell are used considering the current goal? Express 1N=1/g where g=9.82 ms/s^2
Line Did the rater/participant were familiarized with the experimental procedure? Making a PPT at another body location
Line 109-110 Add demographics of the men
Line 111 IMC?
Line 130 temporal summation issue
Line 144 why choosing an ICC(1,1)? Method?
Did you correct for multiple comparison (t-test)
Results
Report results with the same number of decimals
How are the ICC values interpreted? Add a reference (e.g. Landis and Koch 1977)
Discussion
Add also studies mentioning the use of PPT in rehabilitation studies (e.g. rehabilitation, training or ergonomics interventions)
Line 203 add threshold at the end of the sentence
Line 205 add also information concerning spatial aspects, i.e. information the changes in PPT as a function of location – see also line 228 about temporal summation. Here, the authors could consider adding aspect related to spatial summation too.
Check references for normal/capital letters in the reference list
Figures change to SI units (kPa)
Add also a table with the measured values PPTs, CI and differences. See Walton et al (2011)

·

Basic reporting

Grammatically, the paper is poorly written and it is often challenging to extract the meaning from various sections, particularly the introduction and the discussion. I would suggest having someone review the paper for English fluency prior to resubmission.
The flow of the paper is further complicated by a lack of structured organization within each section, making it challenging to follow.

The introduction should be the clearest part of a scientific article and unfortunately this paper's introduction is very confusing. The reader should be presented with thorough background information as well as a clear description of what is being studied in the present paper. The fact that a pressure algometer was used on both a load cell and force platform is hinted at but is not made clear in the introduction, and in fact it does not become clear until reading the materials and methods. In addition, there is no clear description of how the adapted portable hanging scale is different from currently used pressure algometers. Some of the content of the introduction should be condensed into one sentence to state the potential use of it in low income countries instead of mentioning this multiple times. Pressure pain threshold (PPT) is explained well in the beginning of the introduction.

There is inconsistency in the manner in which authors of other studies are cited throughout the paper, especially in the discussion.

Experimental design

The study is very interesting, and indeed, has the potential for a real clinical impact.

Unfortunately, the language overcomplicates the description of the study design. I am not sure what is the relevance of reporting the how people were trained on non-consecutive days/hours?

Specific comments in attachment.

Validity of the findings

The hypothesis stated in the discussion is not the same as the hypothesis used in the introduction. Please clarify.

The authors should explain clearly the differences between commercially used PA and the equipment they used in this study ( adapted PA?). In addition, it is important that they are consistent in referring to their equipment as an adapted PA as in the caption of Figure 1. There are multiple times in this paper where the adapted PA is referred to simply as the PA.

Additional comments

This is a well designed study with the potential for clinical impact. The problems are not with the methodology, but with the writing and presentation of the findings. Although this paper requires significant editing to improve the language and overall flow, I believe that will make it much more accessible and easier to follow when reading it.

---

## Round 0.2 · Major Revisions

Dear authors,

The reviewers and this editor are in agreement that this manuscript needs a major revision to be further considered for publication. Each reviewer has listed a number of points that must be addressed before the manuscript is accepted. These points must be addressed substantially and thoroughly. This reviewer hopes that the manuscript can be revised following the requests of both reviewers. Sincerely,

·

Basic reporting

see below

Experimental design

see below

Validity of the findings

see below

Additional comments

The MS has improved but it is difficult to see what was changed... moreover, the MS lacks still of scientific rigor...
here are points that need further attention:
- the authors need to correct erroneous use og unit's abbreviations: Kg/kg - for PPT, it should be kPa not KPa. check the entire paper
- mention how to transform Newtons (N/cm2) or kilograms of force (Kgf/cm2) per squared cm to kPa in the introduction.
- cite other reliability papers for PPT assessment - make an additional search
- force platform. gold standard for what? not clear a load cell can also measure force!
- concerning re-calibration: how often is this necessary? Somedic provide a weight resulting in a 100kPa reading. this is recommended to check every time one use the pressure algometer
- what is meant by compression measurements?

·

Basic reporting

Basic reporting

Overall, the efforts undertaken by the authors to improve this paper appear substantial and it has improved considerably compared to the initial version. However, my primary concern remains that there are frequent and consequential errors in grammar, diction, and syntax that ultimately fail to construct a narrative that is easy to follow. While the structure of the introduction has been improved quite nicely, the discussion remains disjointed and confusing, forcing the reader too often to draw their own conclusions as to why the authors included the examples or references. The “procedures” section is also disorganized. For instance, the section regarding intra- and inter-rater reliability begins by describing which muscle was tested by two raters. It then transitions to describing the training protocol. It then transitions back to describing a third rater, however it is unclear if at this point the authors are still describing the training or have returned to discussing the initial experiment.

Experimental design

Experimental design

My original comments remain true. This is an interesting study with the potential for real clinical impact. The study is well designed. However, it is the description of the study that requires improvement.

Validity of the findings

No comments

Additional comments

Specific comments

Line 36-36 – The P values should be reported (at least in parentheses) in addition to the qualifiers “moderate” and “excellent” or else these appear to be subjective descriptions.

Line 39 – “valid intra-rater reliable measures” does not make sense. It is possible this should read “valid and reliable…” however given the later descriptions in this manuscript, I am not sure that this author’s intent

Line 52 – The word “instrument” is more appropriate than “equipment,” and replace “on” with “for”

Line 65 – Replace “equipment” with instrument.” The mechanism is more appropriately described as “suspended from an attachment” than “in a suspended manner”

Line 70-71 – Confusing description of the validation process. A “score” implies either a standardized or specific rating system, which likely differs depending on what is being tested and in what manner it is being tested.

Line 74 – Incorrect use of comma after “validity” – consider “as well as” or similar

Line 78-79 – Remove “current” as this is implied by the fact that a hypothesis is being discussed in the introduction. This sentence should state “…algometer is valid and reliable to be considered an acceptable method…” or similar. The purpose of this paper is to describe the use of this tool and demonstrate its use – not to set forth a new standard of care/measurement.

Line 106 – Numerals less than 10 should be written (ex. “two” not “2”)

Line 108 – Rephrase as “alcohol consumption within five days prior to…”

Line 113-118 – This is a confusing transition from describing the test to the training protocol (which by definition must have taken place prior to the test being performed). To maintain a structured flow, it is strongly advised that all training descriptions be completed before discussing how the tests were carried out.

Line 122 – “Pieces of information” should be described as “data”

Line 123-124 – Introduction of a third rater at this time is another example of how the flow of this section is harmed by the writing. This should have been described immediately following the description of the first two raters without transitioning first to describe training. The transition in 124 to raters in training further confuses the issue as it becomes even less clear if the author is describing the training process or the experiment at this time

Line 129 – Delete “positioned.” Which sites are being referenced? It is assumed to be the middle deltoid, but this should be stated again

Line 144-146 – Multiple instances of “measures” being used instead of “measurements”

Line 146 – “Validation” should be used in place of “validity”

Line 151-153 – This should be stated in the abstract instead of the subjective terms used

Line 173 – The use of “both” implies two. However, earlier three raters were described. It should be clarified as to which portion of the test this is referring to. In general, the description of the rate training vs. the actual experiment makes interpretation of the results unnecessarily complicated.

Line 198-199 – “…statistically significant results for the expected correlation of both equipment in measuring the same construct” should be rephrased using more straightforward language to emphasize the findings

Line 200 – Replace “has validity and reliability” with “is both valid and reliable enough…”

Line 202 – This is better described as an “acceptable alternative” rather than simply an alternative

Line 208 – Should “prospective” be “objective” – unclear what author’s intent is


Line 209 – Pressure algometry is better described as a diagnostic aid, rather that implied as a primary method of obtaining a diagnosis

Lines 212, 232 – Replace “include” with “utilize”

---

## Round 0.3 · Minor Revisions

Your manuscript is improved, please update the document according to the input from Reviewer 2 and please make the following changes before resubmission.

- Abstract – add the unit after the reported SEM (kPa).
- Abstract – change “A very strong” to “Very strong”
- Abstract – In the phrase “… demonstrated both the validity and …”, delete the word “both”.
- Line 62. Delete the word “readings”
- Line 102. Change “3 digits” to “3-digit”
- Line 307. Change “assemblage” to “assembly”.

·

Basic reporting

This version of the manuscript shows a significant improvement in structure and overall writing. It is clear and much easier to follow than previous versions. There are a few instances in which the literature is referenced in an inconsistent manner (see the attached document for specific lines). These should be addressed prior to publication.

Experimental design

No additional comments.

Validity of the findings

No additional comments

Additional comments

The efforts undertaken to improve the English and to provide clarity and structure to this manuscript are clear and I appreciate the effort put forth. The manuscript has improved dramatically as a result.

---

## Round 0.4 · accepted · Accept

Thank you for revising the manuscript.

During the proof stage, Line 160: 'pressed' should be changed to 'pressure'.